# The Diagnostic Differentiation Challenge in Acute Appendicitis: How to Distinguish between Uncomplicated and Complicated Appendicitis in Adults

**DOI:** 10.3390/diagnostics12071724

**Published:** 2022-07-15

**Authors:** Benedicte Skjold-Ødegaard, Kjetil Søreide

**Affiliations:** 1Department of Surgery, Haugesund Hospital, 5527 Haugesund, Norway; 2Department of Gastrointestinal Surgery, Stavanger University Hospital, 4011 Stavanger, Norway; ksoreide@mac.com; 3Faculty of Health and Medicine, University of Stavanger, 4021 Stavanger, Norway; 4Department of Clinical Medicine, University of Bergen, 5007 Bergen, Norway

**Keywords:** appendicitis, appendectomy, disease severity

## Abstract

(1) Background: How to best define, diagnose and differentiate uncomplicated from complicated acute appendicitis remains debated. Hence, the aim of this review was to present an overview of the current knowledge and emerging field of acute appendicitis with a focus on the diagnostic differentiation of severity currently subject to ongoing investigations. (2) Methods: We conducted a PubMed search using the MeSH terms “appendicitis AND severity” and “appendicitis AND classification”, with a focus on studies calling appendicitis as ‘uncomplicated’ or ‘complicated’. An emphasis on the last 5 years was stressed, with further studies selected for their contribution to the theme. Further studies were retrieved from identified full-text articles and included per the authors’ discretion. (3) Results: The assumption that appendicitis invariably will proceed to perforation has been outdated. Both uncomplicated and complicated appendicitis exist with likely different pathophysiology. Hence, this makes it important to differentiate disease severity. Clinicians must diagnose appendicitis, but, in the next step, also differentiate between uncomplicated and complicated appendicitis in order to allow for management decisions. Diagnostic accuracy without supportive imaging is around 75–80% and, based on clinical judgement and blood tests alone, the negative appendectomy rate has been described as high as 36%. More research is needed on available biomarkers, and the routine use of imaging still remains debated. Scoring systems have the potential to improve diagnostic accuracy, but no scoring system has yet been validated for differentiating disease severity. Currently, no universally agreed definition exists on what constitutes a complicated appendicitis. (4) Conclusions: Uncomplicated and complicated appendicitis appear to have different pathophysiology and should be treated differently. The differentiation between uncomplicated and complicated appendicitis remains a diagnostic challenge.

## 1. Introduction

Acute appendicitis is the most common surgical emergency worldwide. The lifetime risk is reported to be at 7–8% [1], and in the United States alone, approximately 300,000 appendectomies are performed every year. Thus, the societal influences and the healthcare burden from acute appendicitis are considerable [2]. Despite surgery being the standard treatment for over a century, several unresolved issues still pertain to acute appendicitis. The diagnosis remains a challenge with several symptoms and signs that are equivocal of the condition (Figure 1) and may mimic serval other differential diagnoses [3,4]. Moreover, imaging studies increase sensitivity and specificity for the condition, but remains debated due to the resources and radiation exposure involved. More recently, the debate over non-operative management with antibiotics alone has emerged for uncomplicated appendicitis. However, the challenge remains in establishing how to best define, diagnose and differentiate the uncomplicated from complicated acute appendicitis. Hence, the aim of this review was to present an overview of the current knowledge and emerging field of acute appendicitis with a focus on the diagnostic differentiation of severity currently subject to ongoing investigations.

## 2. Materials and Methods

We conducted a narrative review based on a PubMed search using the MeSH terms “appendicitis AND severity” and “appendicitis AND classification”, with a focus on studies defining appendicitis as either ‘uncomplicated’ or ‘complicated’. An emphasis on the last 5 years was stressed, with further studies selected for their contribution to the theme. Further studies were retrieved from identified full-text articles and included per the authors’ discretion. We apologize to those authors’ whose work was not cited due to the limitations in this short report.

## 3. Results

We present a narrative review based on existing knowledge, ongoing debate and available studies.

### 3.1. Emerging Changes in Management from New Understanding of Disease

Historically, a certain rate of negative explorations has been accepted as good surgical practice [5]. However, negative appendectomies come with the risk of postoperative complications, and there have even been reported cases of fatality [6,7]. This has led to the current recognition that a negative appendectomy should not be considered a simple and harmless operation [8], and a low ‘negative appendectomy rate’ (in short, NAR) is increasingly used as a quality indicator for appendicitis management.

When acute appendicitis was first described by Fitz in 1886, this disease was often deadly, and the lack of antibiotics made it essential to rapidly remove the appendix surgically in order to avoid a life-threatening infection, sepsis and death. The effect of surgery to treat appendicitis was one of the early successes of surgery; hence, it has been rather difficult up to modern times to consider any other alternative in management. From the earliest description it was assumed—and this persisted until relatively recently—that acute appendicitis would inevitably proceed to perforation if not removed at a non-perforated state. This is, however, a pathophysiological hypothesis that has never been proven [5]. Epidemiologic trends suggest that the pathophysiology of nonperforated appendicitis may instead differ from that of perforated appendicitis [5]. If the condition involves at least two pathophysiological different disease pathways (e.g., an inflammatory, uncomplicated type and one that may ultimately progress to perforation), there is a need to develop tools to potentially differentiate between these conditions in clinical practice in order to allow for a stratified management in decisions. This has led to a situation where the diagnostic process of acute appendicitis could be considered a two-step process (Figure 1) where the first step is to determine the diagnosis (e.g., whether or not the patient has acute appendicitis), while the second step is to differentiate between the severity (e.g., an uncomplicated and a complicated) of acute appendicitis [9]. Alas, both situations are still hampered by a number of challenges, and both diagnosis and severity determination pose a huge challenge to most health care systems worldwide.

### 3.2. Diagnostic Prevalence of Acute Appendicitis

As many as 7–10% of all admissions to emergency departments (Figure 1) are patients presenting with acute abdominal pain [10]. The clinicians’ first task is to identify the patients where the pain is caused by acute appendicitis. The epidemiology and prevalence of appendicitis is highly related to the population at risk, with age- and sex-specific patterns of presentation. Acute appendicitis is most frequent between the age of 10 and 30, whereas children under 10 years have the lowest incidence [4,11]. The male–female ratio is 1.4:1 [4,11], but, despite this, the risk of undergoing appendectomy is much lower for males than for females (12 vs. 23%) [11]. Men are more likely to have a perforated appendicitis, and the incidence of perforated appendicitis has been increasing [4]. Ethnicity data show that appendicitis is less common in non-white groups, and both environmental factors and genetic effects are thought to play a role in the development of the disease [1].

#### 3.2.1. Clinical Diagnosis

Based on clinical judgement alone, negative appendectomy rates have been reported to be as high as 36% [6,12], and diagnostic accuracy without imaging has been shown to be as low as 75–80% [8,9]. Hence, most believe that a clinical diagnosis alone is insufficient for the diagnosis and even more so to establish a differentiation between uncomplicated and complicated disease [3].

#### 3.2.2. Biomarkers

During the last decades, research has been conducted to identify biomarkers that could differentiate between uncomplicated and complicated appendicitis. Biomarkers have the potential to provide noninvasive objective criteria without any adverse effects on the patient [13]. Serum bilirubin has been suggested as a possible marker of perforation, as hyperbilirubinemia commonly occurs in patients with septic conditions [14,15]. Bacteremia can cause endotoxemia leading to impaired excretion of bilirubin from the bile canaliculi [15]. However, hyperbilirubinemia alone has a low overall accuracy to diagnose a complicated appendicitis with anticipated perforation. In cases where elevated bilirubin occurs, the patient is more likely to be diagnosed with complicated appendicitis [14], and normal bilirubin supports the presence of an uncomplicated appendicitis [15]. More research is needed to evaluate whether measurement of serum bilirubin should be integrated into the diagnostic tools when it comes to appendicitis.

Combining two biomarkers results in the neutrophile-to-lymphocyte ratio, a recent systematic review [16] has proven it to be promising in diagnosing acute appendicitis, but also in differentiating uncomplicated from complicated appendicitis. Other biomarkers, such as PNP (Proportion of Polymorphnuclear) cells, IL-6 and S100A8/A9 (calprotectin), have limited clinical value due to low specificity [6].

In order to be a useful biomarker, the cost and real-time feasibility must also be taken into account. More research is needed when it comes to biomarkers; it is probably more likely than one or more biomarkers should be incorporated into the clinical scoring systems rather than be used independently. Precision medicine techniques also have the potential to add to the strength of the diagnostics [17].

#### 3.2.3. Imaging Features

Increased use of imaging has been shown to markedly decrease the negative appendectomy rate, and the introduction of mandatory imaging in the Netherlands with the resulting use of imaging of 99.5% in patients reduced the rate from 23% to 3,2%. In contrast, 32.8% of patients with suspected appendicitis in the UK receive imaging and the negative appendectomy rate is reported to be 20.6% [18] Still, the need for imaging in all patients remain highly debated [3].

Unselected patients with clinical suspicion of acute appendicitis typically have a prevalence of about 25–30%. Even with the high sensitivity and specificity of CT (at about 0.95), a substantial number of false positive will result from the low pretest probability if doing a CT scan on all patients with clinical suspicion. The expert panel of WSES 2020 Jerusalem guidelines highly debated whether patients under 40 years should have mandatory imaging [3]. They then suggested a strategy where scoring systems can be used to stratify patients with suspected appendicitis into low-, intermediate- and high-probability groups. CT imaging should then be performed at the intermediate group. In the high-probability group (pretest greater than 90%) CT could be considered unnecessary, as a negative CT will still come with a posttest probability of acute appendicitis of 30% [3].

This is based on the need to distinguish appendicitis from non-appendicitis. Assuming that uncomplicated and complicated appendicitis have different pathophysiology and, thus, should be treated differently, it is interesting whether imaging can safely differentiate the two. There has been a considerable failure rate at approximately 30% associated with non-operative treatment of presumed uncomplicated appendicitis. It has been speculated whether it is actually the limited diagnostic performance of CT in differentiating complicated from uncomplicated appendicitis that may be the cause of this failure rate [19]. A retrospective study from 2019 [19] found a pooled sensitivity using gestalt assessment at 64%. This would suggest that a significant portion of the patients where CT concluded with uncomplicated appendicitis in fact would have a complicated appendicitis and thus be at risk for failure of conservative treatment. Therefore, despite the high sensitivity of CT in diagnosing acute appendicitis (0.95), the ability to distinguish between uncomplicated and complicated appendicitis seems to be far lower, with a sensitivity around 0.64. As there exists different treatment options for uncomplicated appendicitis, an accurate categorization of appendicitis is very important, and standardized CT criteria for severity should be established [20].

A systematic review [21] concluded that patients diagnosed with appendicitis had significantly larger outer diameter of the vermiform appendix than patients without appendicitis did. Complicated appendicitis was associated with significantly larger outer diameter (13.4 mm, 95% CI 12.2–14.6) when compared to uncomplicated appendicitis (10.1 mm, 95% CI 9.5–10.8). This is based on prospective data and it has, to our knowledge, not been proven in clinical trials.

A recent review [4] defines CT-findings of uncomplicated appendicitis to include dilated appendix (≥7 mm); appendiceal wall thickening, hyperenhancement, or both; and inflammatory stranding of the periappendiceal fat tissue. The presence of gas within the appendiceal lumen generally suggests patency with the coecum and excludes appendicitis. Complicated appendicitis can be recognized by extraluminal appendicolith, abscess formation, appendiceal wall defect, extraluminal gas, ileus, periappendiceal or free intraperitoneal fluid, and severe periappendiceal inflammation or phlegmon.

No imaging modality can still reliably rule out a complicated presentation of acute appendicitis [9,22], and, therefore, no safe differentiation between uncomplicated and complicated appendicitis can be made upon imaging alone [23]

#### 3.2.4. Scoring Systems

Risk stratification using clinical scoring systems could have the potential to improve diagnostics. There exist multiple such scoring systems, including the Alvarado score [24] and the Appendicitis Inflammatory Response (AIR) score [25,26]. Both scoring systems are validated, but recently the AIR score has outperformed the Alvarado score [26,27,28]. The ability of the scoring systems to accurately rule out appendicitis was equal to the clinical judgement of a senior surgeon [14]. However, the scoring systems are validated from differentiating the patients with a low risk of appendicitis from the patients with a high risk, and, thus, have the potential to reduce admissions to the hospital, the use of diagnostic imaging, unnecessary surgery and costs compared to standard judgement [26]. Neither of these scoring systems are validated to distinguish between uncomplicated and complicated appendicitis [14].

Atema et al. [29] developed scoring systems based on clinical features (age, duration of symptoms, body temperature, WBC count and CRP), CT features (destruction of the appendiceal wall, extraluminal free air, periappendiceal fluid and presence of an appendicolith) and ultrasound features (destruction of the appendiceal wall, periappendicular fluid, presence of an appendicolith and periappendiceal fat infiltration). One scoring system was based on CT features, and one scoring system was based on ultrasound features. The authors found that with use of these scoring systems, a substantial group of patients with a low probability of complicated appendicitis or alternative complicated disease (5 per cent) can be identified.

Recently two studies have described scoring systems combining clinical and imaging features to differentiate between uncomplicated and complicated appendicitis [9]; the Scoring systems for Appendicitis Severity (SAS) and the APpendicitis Severity Index (APSI). None of them have yet been validated externally.

#### 3.2.5. Classification

Despite the clinical importance of distinguishing complicated from uncomplicated appendicitis, no universally agreed definition exists on how to classify the two. Several attempts have been made to make definitions (Table 1).

The American Association for the Surgery of Trauma (AAST) developed a grading system to provide a uniform method to assess disease severity, and uses clinical, radiographic, operative, and pathological criteria to assign a score of 1 to 5, where 1 is least severe and 5 is most severe [30]. This scoring system has been validated and has the potential to serve as a useful benchmarking measure by allowing the comparison of patients according to objective measures of disease severity [30].

**Table 1 diagnostics-12-01724-t001:** Proposed definition of uncomplicated/complicated appendicitis.

		Proposed Definition
EAES [31]	European Association of Emergency Surgery	Complicated appendicitis: Gangrenous appendicitis with or without perforation, appendicitis with an intraabdominal abscess, and appendicitis with periappendicular contained phlegmon or purulent/free fluid
WSES [11,12]	World Society of Emergency Surgery	“Complicated appendicitis … the common component of perforation, it may or may not also include non-perforated gangrenous AA, the presence of a fecalith and/or AA in the presence of pus, or purulent peritonitis, or abscess”
Moris et al. [4]	JAMA review	Uncomplicated appendicitis: AA without clinical or radiographic signs of perforation (inflammatory mass, phlegmon, or abscess). Complicated appendicitis: Appendiceal rupture with subsequent abscess or phlegmon formation.
Flum et al. [10]	The CODA collaborative	Complicated appendicitis: Septic shock, diffuse peritonitis, recurrent appendicitis, evidence of severe phlegmon on imaging, walled-off abscess, free air or more than minimal free fluid, or evidence suggestive of neoplasm

### 3.3. Changing Treatment with Evolving Views on Appendicitis as A Disease

The aim of the diagnostics should be to safely and accurately differentiate disease severity in order for identify the patients that should be taken directly to surgery, and the patients that could be threated effectively with antibiotics alone [1], or even resolve spontaneously [9,32]. Successful use of antibiotic therapy to treat acute appendicitis was reported more than 60 years ago [33], and in the last two decades, there has been an increased focus on non-operative management of acute appendicitis.

Uncomplicated and complicated appendicitis have been shown to differ in mortality as well as morbidity. The morbidity risk is reported to be 6.9% in uncomplicated appendicitis, but rises to 20.1% in complicated cases [12] Growing evidence, however, supports that perforation does not happen inevitably in all patients with acute appendicitis [34], and being able to safely identify the uncomplicated cases will have great impact on treatment options, as the majority of patients with uncomplicated acute appendicitis can be treated with an antibiotic-first approach [10]. The triad of C-reactive protein level below 60 mg/L, white blood cell count lower than 12 × 10^9^ and age younger than 60 years has been associated with antibiotic treatment success [1].

Many trials have found that the incidence of major complications are 2- to 4-fold higher in antibiotic-treated patients compared to appendectomy-treated patients [4]. A large, randomized trial [33] found that antibiotics were noninferior to appendectomy on the basis on 30-day health status. Complications were more common in the antibiotics group than in the appendectomy group but were attributable to patients with an appendicolith, who additionally appeared to have a higher risk of serious adverse events. Presence of an appendicolith has, in several studies, been identified as an important risk factor for antibiotic treatment failure [4]. There is, however, a substantial risk of needing a subsequent appendectomy with antibiotic treatment, and appendectomy was in the CODA collaborative study performed in 11% of the participants in 48 h, in 20% by 30 days, and in 29% by 90 days [33]. Long-term follow up showed that the percentage of patients who underwent subsequent appendectomy was 40% after 2 years [35]. This is consistent with earlier findings that approximately 60% of adult patients with acute uncomplicated appendicitis can be treated successfully with antibiotics [4].

The APPAC trial [36] failed to establish the non-inferiority of antibiotic treatment, but conservative management is still considered a safe treatment option in patients with verified uncomplicated appendicitis [10,35,37]. However, there is a considerable proportion (up to 40%) of the patients treated conservatively that will later need an appendectomy, and the patients should be informed of this. A recent meta-analysis [18] finds that antibiotic treatment comes with a significantly lower treatment success rate, but both length of stay and duration of symptoms are not different in patients treated with antibiotics compared to those who undergo surgical treatment.

## 4. Discussion

The “uncomplicated” appendicitis comes with a dilemma. Differentiating the treatment for uncomplicated and complicated appendicitis also implies the need to safely differentiate the two before treatment can be started. No universally agreed classification exists, and neither use of clinical diagnosis, biomarkers, imaging nor scoring systems have proved sufficiently efficient in differentiating uncomplicated from complicated appendicitis. Not missing out on the diagnosis of complicated appendicitis should be considered vital even in the era of modern medicine as a perforated appendicitis still in modern medicine carries a considerable risk of morbidity and even a not insignificant risk of mortality.

Uncritical use of imaging comes with the risk of diagnosing mild cases that would resolve spontaneously without treatment—with following unnecessary surgery and/or overuse of antibiotics.

The art of balancing over-diagnostics against the potential dangers of missing the diagnosis of complicated appendicitis is still a challenge for clinicians. Further research is needed to develop diagnostic tools necessary to safely differentiate between uncomplicated and complicated appendicitis, with focus on indicators that can predict the success of non-operative management.

## 5. Conclusions

Nonoperative management is considered a safe treatment for patients with uncomplicated appendicitis [4,18]. However, how to safely differentiate an uncomplicated appendicitis from a complicated remains a diagnostic and semantic challenge as there are no uniform standards or universally agreed guidelines for this. How to distinguish between uncomplicated and complicated appendicitis should be one of the focus areas in future research of appendicitis, with the aim of providing clinicians with diagnostic guidelines.

## Figures and Tables

**Figure 1 diagnostics-12-01724-f001:**
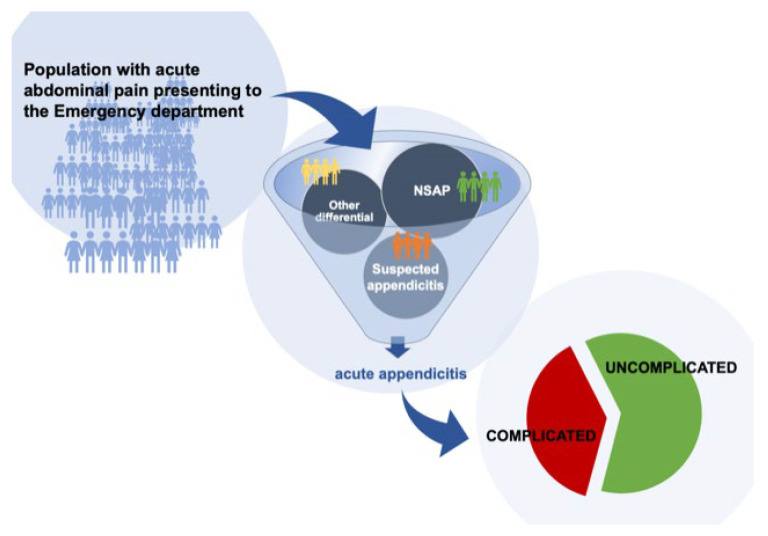
NSAP = Non-Specific Abdominal Pain.

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
