# Peer review of "The Diagnostic Differentiation Challenge in Acute Appendicitis: How to Distinguish between Uncomplicated and Complicated Appendicitis in Adults"

_diagnostics, 2022, doi:10.3390/diagnostics12071724_

Round 1

Reviewer 1 Report

Reviewer's Comments to Authors:

1. Discriminating complicated from uncomplicated appendicitis: Epidemiological trends suggest that the pathophysiology of non-perforated appendicitis may differ from perforated appendicitis.

According to the difference between the above two diagnoses and the follow-up and treatment are also different, whether to providing further standard reference data can provide clinicians with diagnostic reference guidelines?

2. After an in-depth literature review of this manuscript, based on the author's experience, for the diagnostic accuracy (to rule in or rule out), diagnostic tests for acute appendicitis, clinical opinion, laboratory data, clinical scoring system (Appendicitis Severity Score), and imaging analysis, are there any unique recommendations or are more the perspective of precision medicine?

 3. Future research on appendicitis should include a strict focus on the difficult aspects of distinguishing between complex and non-complex diseases. Could you briefly provide your personal recommendations after reviewing the literature?

 4.  Although this manuscript is not a novel review of an article, it also provides some valuable references for clinicians and related patient safety and diagnosis and treatment. According to the author's point of view, can you list the emphasized learning objectives and innovative ideas? To provide further advice and perspectives unique to surgeons?

Author Response

  1. Discriminating complicated from uncomplicated appendicitis: Epidemiological trends suggest that the pathophysiology of non-perforated appendicitis may differ from perforated appendicitis.

According to the difference between the above two diagnoses and the follow-up and treatment are also different, whether to providing further standard reference data can provide clinicians with diagnostic reference guidelines?

RE: We agree that providing clinicians with dagnostic guidelines should be a priority and have emphasized this in the conclusion in the revised version

  1. After an in-depth literature review of this manuscript, based on the author's experience, for the diagnostic accuracy (to rule in or rule out), diagnostic tests for acute appendicitis, clinical opinion, laboratory data, clinical scoring system (Appendicitis Severity Score), and imaging analysis, are there any unique recommendations or are more the perspective of precision medicine?

RE: In the revised version, we have briefly added the perspective of precision medicine under 3.2.2.Biomarkers

  1. Future research on appendicitis should include a strict focus on the difficult aspects of distinguishing between complex and non-complex diseases. Could you briefly provide your personal recommendations after reviewing the literature?

RE: We have added the importance of providing clinical guidelines and reformulated “stringent focus” to “one of the focus areas” in order to recognize that there exist several potential research areas on appendicitis

  1. Although this manuscript is not a novel review of an article, it also provides some valuable references for clinicians and related patient safety and diagnosis and treatment. According to the author's point of view, can you list the emphasized learning objectives and innovative ideas? To provide further advice and perspectives unique to surgeons?

RE: By adding the importance of providing clinical guidelines in the conclusion we hope to have made this important learning objective more clear

Reviewer 2 Report

After carefully review the manuscript I think that is suitable for publication in the present form. Congratulations!!!.

Author Response

After carefully review the manuscript I think that is suitable for publication in the present form. Congratulations!!!.

RE: We thank the reviewer for the positive feedback

Reviewer 3 Report

A decent review 

Please add the following review in you discussion as it might add to the manuscript 

Author Response

A decent review

RE: We thank the reviewer for the positive feedback